# Mind the Gap: A Generative Approach to Interpretable Feature Selection and Extraction

**Been Kim**          **Julie Shah**
Massachusetts Institute of Technology
Cambridge, MA 02139
{beenkim, julie_a_shah}@csail.mit.edu

**Finale Doshi-Velez**
Harvard University
Cambridge, MA 02138
finale@seas.harvard.edu

## Abstract

We present the Mind the Gap Model (MGM), an approach for interpretable feature extraction and selection. By placing interpretability criteria directly into the model, we allow for the model to both optimize parameters related to interpretability and to directly report a global set of distinguishable dimensions to assist with further data exploration and hypothesis generation. MGM extracts distinguishing features on real-world datasets of animal features, recipes ingredients, and disease co-occurrence. It also maintains or improves performance when compared to related approaches. We perform a user study with domain experts to show the MGM's ability to help with dataset exploration.

## 1 Introduction

Not only are our data growing in volume and dimensionality, but the understanding that we wish to gain from them is increasingly sophisticated. For example, an educator might wish to know what features characterize different clusters of assignments to provide in-class feedback tailored to each student's needs. A clinical researcher might apply a clustering algorithm to his patient cohort, and then wish to understand what sets of symptoms distinguish clusters to assist in performing a differential diagnosis. More broadly, researchers often perform clustering as a tool for data exploration and hypothesis generation. In these situations, the domain expert's goal is to understand what features characterize a cluster, and what features distinguish between clusters.

Objectives such as data exploration present unique challenges and opportunities for problems in unsupervised learning. While in more typical scenarios, the discovered latent structures are simply required for some downstream task—such as features for a supervised prediction problem—in data exploration, the model must provide information to a domain expert in a form that they can readily interpret. It is not sufficient to simply list what observations are part of which cluster; one must also be able to explain why the data partition in that particular way. These explanations must necessarily be succinct, as people are limited in the number of cognitive entities that they can process at one time [1].

The de-facto standard for summarizing clusters (and other latent factor representations) is to list the most probable features of each factor. For example, top-N word lists are the de-facto standard for presenting topics from topic models [2]; principle component vectors in PCA are usually described by a list of dimensions with the largest magnitude values for the components with the largest magnitude eigenvalues. Sparsity-inducing versions of these models [3, 4, 5, 6] make this goal more explicit by trying to limit the number of non-zero values in each factor. Other works make these descriptions more intuitive by deriving disjunctive normal form (DNF) expressions for each cluster [7] or learning a set of important features and examples that characterizes each cluster [8]. While these approaches might effectively *characterize* each cluster, they do not provide information about

what *distinguishes* clusters from each other. Understanding these differences is important in many situations—such when performing a differential diagnosis and computing relative risks [9, 10].

Techniques that combine variable selection and clustering assist in finding dimensions that distinguish—rather than simply characterize—the clusters [11, 12]. Variable extraction methods, such as PCA, project the data into a smaller number of dimensions and perform clustering there. In contrast, variable selection methods choose a small number of dimensions to retain. Within variable selection approaches, filter methods (e.g. [13, 14, 15]) first select important dimensions and then cluster based on those. Wrapper methods (e.g. [16]) iterate between selecting dimensions and clustering to maximize a clustering objective. Embedded methods (e.g. [17, 18, 19]) combine variable selection and clustering into one objective. All of these approaches identify a small subset of dimensions that can be used to form a clustering that is as good as (or better than) using all the dimensions. A primary motivation for identifying this small subset is that one can then accurately cluster future data with many fewer measurements per observation. However, identifying a minimal set of distinguishing dimensions is the opposite of what is required in data exploration and hypothesis generation tasks. Here, the researcher desires a comprehensive set of distinguishing dimensions to better understand the important patterns in the data.

In this work, we present a generative approach for discovering a global set of distinguishable dimensions when clustering high-dimensional data. Our goal is to find a comprehensive set of distinguishing dimensions to assist with further data exploration and hypothesis generation, rather than a few dimensions that will distinguish the clusters. We use an embedded approach that incorporates interpretability criteria directly into the model. First, we use a logic-based feature extraction technique to consolidate dimensions into easily-interpreted groups. Second, we define important groups as ones having multi-modal parameter values—that is, groups that have *gap* in their parameter values across clusters. By building these human-oriented interpretability criteria directly into the model, we can easily report back what an extracted set of features means (by its logical formula) and what sets of features distinguish one cluster from another without any ad-hoc post-hoc analysis.

## 2  Model

We consider a data-set $\{w_{nd}\}$ with $N$ observations and $D$ binary dimensions. Our goal is to decompose these $N$ observations into $K$ clusters while simultaneously returning a comprehensive list of what sets of dimensions $d$ are important for distinguishing between the clusters.

MGM has two core elements which perform interpretable feature extraction and selection. At the feature extraction stage, features are grouped together by logical formulas, which are easily interpreted by people [20, 21], allowing some dimensionality reduction while maintaining interpretability. Next, we select features for which there is a large separation—or a *gap*—in parameter values. From personal communication with domain experts across several domains, we observed that separation—rather than simply variation—is often as aspect of interest as it provides an unambiguous way to discriminate between clusters.

We focus on binary-valued data. Our feature extraction step involves consolidating dimensions into groups. We posit that there an infinite number of groups $g$, and a multinomial latent variable $l_d$ that indicates the group to which dimension $d$ belongs. Each group $g$ is characterized by a latent variable $f_g$ which contains the formula associated with the group $g$. In this work, we only consider the formulas $f_g = \mathsf{or}$, $f_g = \mathsf{and}$ and constrain each dimension to belong to only one group. Simple Boolean operations like $\mathsf{or}$ and $\mathsf{and}$ are easy to interpret by people. Requiring each dimension to be part of only one group avoid having to solve a (possibly NP-complete) satisfiability problem as part of the generative procedure.

Feature selection is performed through a binary latent variable $y_g$ which indicates whether each group $g$ is important for distinguishing clusters. If a group is important ($y_g = 1$), then the probability $\beta_{gk}$ that group $g$ is present in an observation from cluster $k$ is drawn from a bi-modal distribution (modeled as a mixture of Beta distributions). If the group is unimportant ($y_g = 0$), the the probability $\beta_{gk}$ is drawn from a uni-modal distribution. While a uni-modal distribution with high variance can also produce both low and high values for the probability $\beta_{gk}$, it will also produce intermediate values. However, draws from the bi-modal distribution will have a clear *gap* between low and high values. This definition of important distributions is distinct from the criterion in [17], where

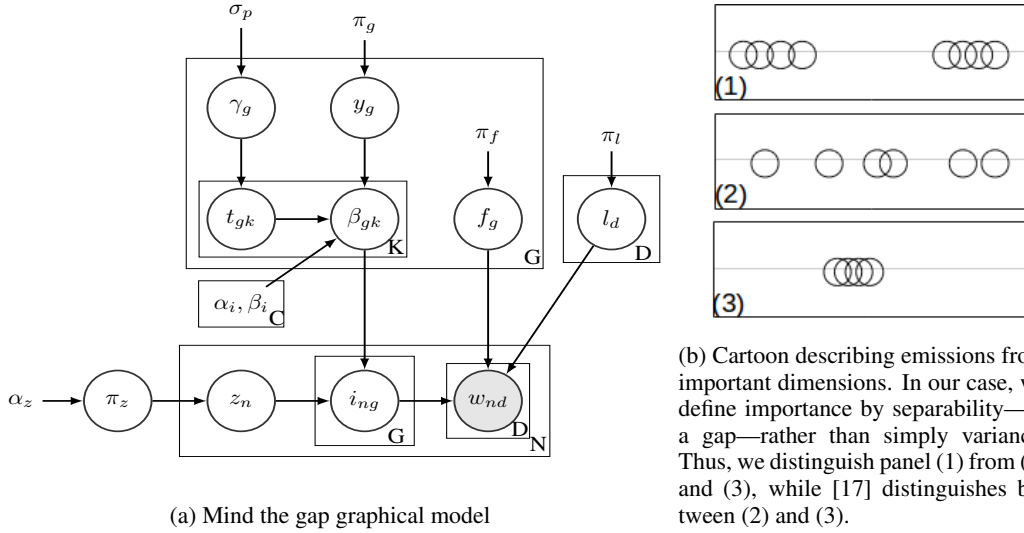

(a) Mind the gap graphical model

(b) Cartoon describing emissions from important dimensions. In our case, we define importance by separability—or a gap—rather than simply variance. Thus, we distinguish panel (1) from (2) and (3), while [17] distinguishes between (2) and (3).

Figure 1: Graphical model of MGM, Cartoon of distinguishing dimensions.

parameters for important distributions were selected from a uni-modal distribution and parameters for unimportant dimensions were shared across all clusters. Figure 1b illustrates this difference.

**Generative Model** The graphical model for MGM is shown in Figure 1. We assume that there are an infinite number of possible groups $g$, each with an associated formula $f_g$. Each dimension $d$ belongs to a group $g$, as indicated by $l_d$. We also posit that there are a set of latent clusters $k$, each with emission characteristics described below. The latent variable $\beta_{gk}$ corresponds to the probability that group $g$ is present in the data, and is drawn with a uni-modal or bi-modal distribution governed by the parameters $\{\gamma_g, y_g, t_{gk}\}$. Each observation $n$ belongs to exactly one latent cluster $k$, indicated by $z_n$. The binary variable $i_{ng}$ indicates whether group $g$ is present in observation $n$. Finally, the probability of some observation $w_{nd} = 1$ depends on whether its associated group $g$ (indicated by $l_d$) is present in the data (indicated by $i_{ng}$) and the associated formula $f_g$.

The complete generative process first involves assigning dimensions $d$ to groups, choosing the formula $f_g$ associated with each group, and deciding whether each group $g$ is important:

$$\pi_l \sim \mathsf{DP}(\alpha_l) \qquad \pi_f \sim \mathsf{Dirichlet}(\alpha_f) \qquad l_d \sim \mathsf{Multinomial}(\pi_l)$$
$$y_g \sim \mathsf{Bernoulli}(\pi_g) \qquad \gamma_g \sim \mathsf{Beta}(\sigma_1, \sigma_2) \qquad f_g \sim \mathsf{Multinomial}(\pi_f)$$

where DP is the Dirichlet process. Thus, there are an infinite number of potential groups; however, given a finite number of dimensions, only a finite number of groups can be present in the data. Next, emission parameters are selected for each cluster $k$:

$$\begin{aligned}
&\mathsf{If}(y_g = 0) &&&& \beta_{gk} \sim \mathsf{Beta}(\alpha_u, \beta_u)\\
&\mathsf{Else}: \quad t_{gk} \sim \mathsf{Bernoulli}(\gamma_g) && \mathsf{If}: \quad t_{gk} = 0: && \beta_{gk} \sim \mathsf{Beta}(\alpha_b, \beta_b)\\
& && \mathsf{Else}: && \beta_{gk} \sim \mathsf{Beta}(\alpha_t, \beta_t)
\end{aligned}$$

Finally, observations $w_{nd}$ are generated:

$$\pi_z \sim \mathsf{Dirichlet}(\alpha_z) \qquad z_n \sim \mathsf{Multinomial}(\pi_z) \qquad i_{ng} \sim \mathsf{Bernoulli}(\beta_{gk})$$
$$\mathsf{If}: \quad i_{ng} = 0: \{w_{nd}|l_d = g\} = 0 \quad \mathsf{Else}: \{w_{nd}|l_d = g\} \sim \mathsf{Formula}_{f_g}$$

The above equations indicate that if $i_{ng} = 0$, that is, group $g$ is not present in the observation, then in that observation, all $w_{nd}$ such that $l_d = g$ are also absent (i.e. $w_{nd} = 0$). If the group $g$ is present ($i_{ng} = 1$) and the group formula $f_g =$ and, then all the dimensions associated with that dimension are present (i.e. $w_{nd} = 1$). Finally, if the group $g$ is present ($i_{ng} = 1$) and the group formula $f_g =$ or, then we sample the associated $w_{nd}$ from all possible configurations of $w_{nd}$ such that at least one $w_{nd} = 1$.

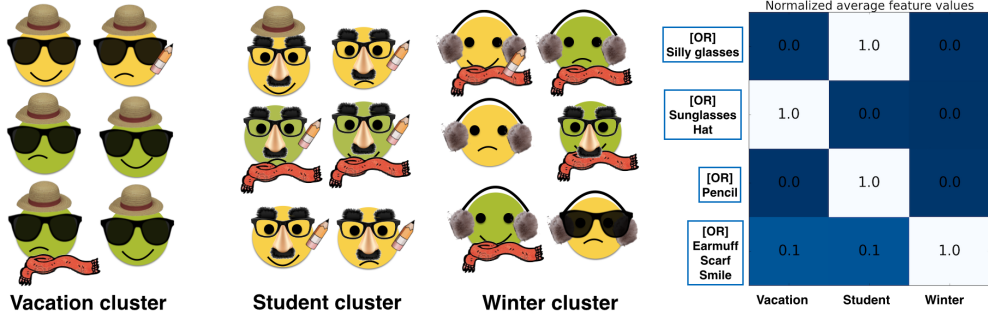

Figure 2: Motivating examples with cartoons from three clusters (vacation, student, winter) and the distinguishable dimensions discovered by the MGM.

Let $\theta = \{y_g, \gamma_g, t_{gk}, \beta_{gk}, l_d, f_g, z_n, i_{ng}\}$ be the set of variables in the MGM. Given a set of observations $\{w_{nd}\}$, the posterior over $\theta$ factors as

$$Pr(\{y_g, \gamma_g, t_{gk}, \beta_{gk}, l_d, f_g, z_n, i_{ng}\}|\{w_{nd}\}) = \prod_g^G p(y_g|\rho)p(\gamma_g|\sigma)p(f_g|\alpha)\cdot$$

$$[\prod_k^K p(t_{gk}|\gamma_g)p(\beta_{gk}|t_{gk}, y_g)]p(\kappa|\alpha)\prod_d^D p(l_d|\kappa)p(\pi|\alpha)\prod_n^N p(z_n|\pi)$$

$$\prod_n^N\prod_g^G p(i_{ng}|\beta, z_n)\prod_n^N\prod_d^D p(w_{nd}|i_{ng}, f, l_d)] \tag{1}$$

Most of these terms are straight-forward to compute given the generative model. The likelihood term $p(w_{nd}|i_{ng}, f, l_d)$ can be expanded as

$$p(w_{n\cdot}|i_{ng}, f, l_d) = \prod_{d,g}[(0)^{\mathbf{1}(i_{ng}=1)(1-\mathbf{SAT}(g;w_{n\cdot},f_g,l_d))}(1)^{\mathbf{1}(i_{ng}=1)\mathbf{SAT}(g;w_{n\cdot},f_g,l_d)}$$

$$(0)^{\mathbf{1}(i_{ng}=0)\mathbf{1}(l_d=g)\mathbf{1}(w_{nd}=1)}(1)^{\mathbf{1}(i_{ng}=0)\mathbf{1}(l_d=g)\mathbf{1}(w_{nd}=0)} \tag{2}$$

where we use $w_{n\cdot}$ to indicate the vector of measurements associated with observation $n$. The function $\mathbf{SAT}(g; w_{n\cdot}, f_g, l_d)$ indicates whether the associated formula, $f_g$ is satisfied, where $f_g$ involves $d$ dimensions of $w_{n\cdot}$ that belong to group $l_d$.

**Motivating Example** Here we provide an example to illustrate the properties of MGM on a synthetic data-set of 400 cartoon faces. Each cartoon face can be described by eight features: earmuffs, scarf, hat, sunglasses, pencil, silly glasses, face color, mouth shape (see Figure 2). The cartoon faces belong to three clusters. Winter faces tend to have earmuffs and scarves. Student faces tend to have silly glasses and pencils. Vacation faces tend to have hats and sunglasses. Face color does not distinguish between the different clusters.

The MGM discovers four distinguishing sets of features: the vacation cluster has hat or sunglasses, the winter cluster has earmuffs or scarfs or smile, and the student cluster has silly glasses as well as pencils. Face color does not appear because it does not distinguish between the groups. However, we do identify both hats and sunglasses as important, even though only one of those two features is important for distinguishing the vacation cluster from the other clusters: our model aims to find a comprehensive list the distinguishing features for a human expert to later review for interesting patterns, not a minimal subset for classification. By consolidating features—such as (sunglasses or hat)—we still provide a compact summary of the ways in which the clusters can be distinguished.

## 3 Inference

Solving Equation 1 is computationally intractable. We use variational approach to approximate the true posterior distribution $p(y_g, \gamma_g, t_{gk}, \beta_{gk}, l_d, f_g, z_n, i_{ng} | \{w_{nd}\})$ with a factored distribution:

$$q_{\eta_g}(y_g) \sim \mathsf{Bernoulli}(\eta_g) \qquad q_{\lambda_{gk}}(t_{gk}) \sim \mathsf{Bernoulli}(\lambda_{gk})$$

$$q_{\ell_g}(\gamma_g) \sim \mathsf{Beta}(\ell_{g1}, \ell_{g2}) \qquad q_{\phi_{gk}}(\beta_{gk}) \sim \mathsf{Beta}(\phi_{gk1}, \phi_{gk2})$$

$$q_{\tau_n}(\pi) \sim \mathsf{Dirichlet}(\tau) \qquad q_{\nu_n}(z_n) \sim \mathsf{Multinomial}(\nu_n) \qquad q_{i_{ng}}(i_{ng}) \sim \mathsf{Bernoulli}(o_{ng})$$

$$q_{c_d}(l_d) \sim \mathsf{Multinomial}(c_d) \qquad q_{e_g}(f_g) \sim \mathsf{Bernoulli}(e_g)$$

where in addition we use a weak-limit approximation to the Dirichlet process to approximate the distribution over group assignments $l_d$. Minimizing the Kullback-Leibler divergence between the true posterior $p(\theta | \{w_{nd}\})$ and the variational distribution $q(\theta)$ corresponds to maximizing the evidence lower bound (the ELBO) $E_q[\log p(\theta | \{w_{nd}\})] - H(q)$ where $H(q)$ is the entropy.

Because of the conjugate exponential family terms, most of the expressions in the ELBO are straightforward to compute. The most challenging part is determining how to optimize the variational terms $q(l_d), q(i_{ng})$, and $q(f_g)$ that are involved in the likelihood in Equation 2. Here, we first relax our generative process of **or** to have it correspond to independently sampling each $w_{nd}$ with some probability $s$. Thus, Equation 2 becomes

$$p(w_n. | i_{ng}, f_g, l_d) = \prod_{d,g} [(0)^{\mathbf{1}(f_g=\mathsf{and})\mathbf{1}(l_d=g)\mathbf{1}(i_{ng}=1)\mathbf{1}(w_{nd}=0)} (1)^{\mathbf{1}(f_g=\mathsf{and})\mathbf{1}(l_d=g)\mathbf{1}(i_{ng}=1)\mathbf{1}(w_{nd}=1)}$$

$$(1-s)^{\mathbf{1}(f_g=\mathsf{or})\mathbf{1}(l_d=g)\mathbf{1}(i_{ng}=1)\mathbf{1}(w_{nd}=0)} (s)^{\mathbf{1}(f_g=\mathsf{or})\mathbf{1}(l_d=g)\mathbf{1}(i_{ng}=1)\mathbf{1}(w_{nd}=1)}$$

$$(0)^{\mathbf{1}(i_{ng}=0)\mathbf{1}(l_d=g)\mathbf{1}(w_{nd}=1)} (1)^{\mathbf{1}(i_{ng}=0)\mathbf{1}(l_d=g)\mathbf{1}(w_{nd}=0)} \tag{3}$$

With this relaxation, the expression for the entire evidence lower bound is straight-forward to compute. (The full derivations are given in the supplementary materials.)

However, the logical formulas in Equation 3 still impose hard, combinatorial constraints on settings of the variables $\{i_{ng}, f_g, l_d\}$ that are associated with the logical formulas. Specifically, if the values for the formula choice $\{f_g\}$ and group assignments $\{l_d\}$ are fixed, then the value of $i_{ng}$ is also fixed because the feature extraction step is deterministic. Once $i_{ng}$ is fixed, however, the relationships between all the other variables are conjugate in the exponential family. Therefore, we alternate our inference between the extraction-related variables $\{i_{ng}, f_g, l_d\}$ and the selection-related variables $\{y_g, \gamma_g, t_{gk}, \beta_{gk}, z_n\}$.

**Feature Extraction** We consider only degenerate distributions $q(i_{ng}), q(f_g), q(l_d)$ that put mass on only one setting of the variables. Note that this is still a valid setting for the variational inference as fixing values for $i_{ng}, f_g$, and $l_d$, which corresponds to a degenerate Beta or Dirichlet prior, only means that we are further limiting our set of variational distributions. Not fully optimizing a lower bound due to this constraint can only lower the lower bound.

We perform an agglomerative procedure for assigning dimensions to groups. We begin our search with each dimension $d$ assigned to its own formula $l_d = d$, $f_d = $ or. Merges of groups are explored using a combination of data-driven and random proposals, in which we also explore changing the formula assignment of the group. For the data-driven proposals, we use an initial run of a vanilla k-means clustering algorithm to test whether combining two more groups results in an extracted feature that has high variance. At each iteration, we compute the ELBO for non-overlapping subsets of these proposals and choose the agglomeration with the highest ELBO.

**Feature Selection** Given a particular setting of the extraction variables $\{i_{ng}, f_g, l_d\}$, the remaining variables $\{y_g, \gamma_g, t_{gk}, \beta_{gk}, z_n\}$ are all in the exponential family. The corresponding posterior distributions $q(y_g), q(\gamma_g), q(t_{gk}), q(\beta_{gk})$, and $q(z_n)$ can be optimized via coordinate ascent [22].

## 4 Results

We applied our MGM to both standard benchmark and more interesting data sets. In all cases, we ran 5 restarts of the MGM. Inference was run for 40 iterations or until the ELBO improved by less than 0.1 relative to the previous iteration. Twenty possible merges were explored in each iteration;

|  | MGM | Kmeans | HFS(G) | Law | DPM | HFS(L) | Cc |
|---|---|---|---|---|---|---|---|
| Faces | 0.59 (13) | 0.46 (4) | 0.627 (16) | 0.454 (4) | 0.481 (12) | 0.569 (12) | 0.547 (4) |
| Digits | 0.53 (13) | 0.45 (13) | 0.258 (13) | 0.254 (6) | 0.176 (5) | 0.354 (11) | 0.364 (10) |

Table 1: Mutual information and number of clusters (in parentheses) for UCI benchmarks. The mutual information is with respect to the true class labels (higher is better). Performance values for HFS(G), Law, DPM, HFS(L), and CC are taken from [17].

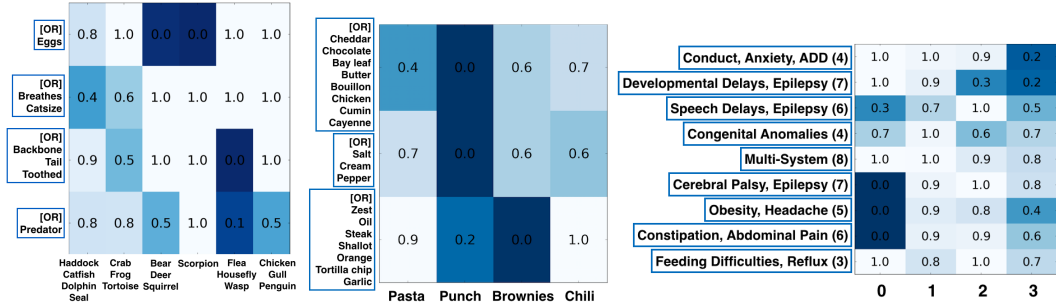

Figure 3: Results on real-world datasets: animal dataset (left), recipe dataset (middle) and disease dataset (right). Each row represents an important feature. Lighter boxes indicate that the feature is likely to be present in the cluster, while darker boxes are unlikely to be present.

each merge exploration involved combining two existing groups into a new group. If we failed to accept our data-driven candidate merge proposals more than three times within an iteration, we switched to random proposals for the remaining proposals. We swept over the number of clusters from K=4 to K=16 and reported the results with the highest ELBO.

## 4.1 Benchmark Problems: MGM discriminates classes

We compared the classification performance of our clustering algorithms on several UCI benchmark problems [23]. The digits data set consists of 11000 16×16 grayscale images, 1100 for each digit. The faces data set consists of 640 32×30 images of 20 people, with 32 images of each person from various angles. In both cases, we binarized the images, setting the value to 0 if the value was less than 128, 1 if the value was greater than 128. These two data-sets are chosen as they are discrete and we have the same versions for comparison to results cited in [17].

The mutual information between our discovered clusters and the true classes in the data sets is shown in Table 1. A higher mutual information between our clustering and known labels is one way to objectively show that our clusters correspond to groups that humans find interesting (i.e. the human-provided classification labels). MGM is second only to HFS(G) in the Faces dataset (second only to HFS(G)) and the highest scoring model in the Digits dataset. It always outperforms k-means.

## 4.2 Demonstrating Interpretability: Real-world Applications

Our quantitative results on the benchmark datasets show that the structure recovered by our approach is consistent with classes defined by human labelers better than or at the level of other clustering approaches. However, the dimensions in the image benchmarks do not have much associated meaning, and the our approach was designed for clustering, not classification. Here, we demonstrate the qualitative advantages of our approach on three more interesting datasets.

**Animals** The animals data set [24] consists of 21 biological and ecological properties of 101 animals (such as "has wings" or "has teeth"). We are also provided class labels such as insects, mammals, and birds. The result of our MGM is shown in Figure 3. Each row is a distinguishable feature; each column is a cluster. Lighter color boxes in Figure 3 indicate that the feature is likely to be present in the cluster, while darker color boxes indicate that the feature is unlikely to be present in the cluster. Below each cluster, a few animals that belong to that cluster are listed.

We first note that, as desired, our model selects features that have large variation in their probabilities across the clusters (rows in Figure 3). Thus, it is straight-forward to read what makes each column different from the others: the mammals in the third column do not lay eggs; the insects in the fifth column are toothless and invertebrates (and therefore have no tails). They are also rarely predators. Unlike the land animals, many of the water animals in columns one and two do not breathe.

**Recipes** The recipes data set consists of ingredients from recipes taken from the computer cooking contest[1]. There are 56 recipes, with 147 total ingredients. The recipes fall into four categories: pasta, chili, brownies or punch. We seek to find ingredients and groups of ingredients that can distinguish different types of recipes. Note: The names for each cluster have been filled in after the analysis, based on the class label of the majority of the observations that were grouped into that cluster.

The MGM distills the 147 ingredients into only 3 important features. The first extracted feature contains several spices, which are present in pasta, brownies, and chili but not in punch. Punch is also distinguished from the other clusters by its lack of basic spices such as salt and pepper (the second extracted feature). The third extracted feature contains a number of savory cooking ingredients such as oil, garlic, and shallots. These are common in the pasta and chili clusters but uncommon in the punch and brownie clusters.

**Diseases** Finally, we consider a data set of patients with autism spectrum disorder (ASD) accumulated over the first 15 years of life [25]. ASDs are a complex disease that is often associated with co-occurring conditions such as seizures and developmental delays. As most patients have very few diagnoses, we limited our analysis to the 184 patients with at least 200 diagnoses and the 58 diagnoses that occurred in at least 5% of the patients. We binarized the count data to 0-1 values.

Our model reduces these 58 dimensions to 9 important sets of features. The extracted features had many more dimensions than in the examples, so we only list two features from each group and provide the total number in parenthesis. Several of the groups of the extracted variables—which did not use any auxiliary information—are similar to those from [25]. In particular, [25] report clusters of patients with epilepsy and cerebral palsy, patients with psychiatric disorders, and patients with gastrointestinal disorders. Using our representation, we can easily see that there appears to be one group of sick patients (cluster 1) for whom all features are likely. We can also see what features distinguish clusters 0, 2, and 3 from each other by which ones are unlikely to be present.

### 4.3 Verifying interpretability: Human subject experiment

We conducted a pilot study to gather more qualitative evaluation of the MGM. We first divided the ASD data into three datasets with random disjoint subsets of approximately 20 dimensions each. For each of these subsets, we prepared the data in three formats: raw patient data (a list of symptoms), clustered results (centroids) from K-means, and clustered results with the MGM with distinguishable sets of features. Both the clustered results were presented as figures such as figure 3 and the raw data were presented in a spreadsheet. Three domain experts were then tasked to explore the different data subsets in each format (so each participant saw all formats and all data subsets) and produce a 2-3 sentence executive summary of each. The different conditions serve as reference points for the subjects to give more qualitative feedback about the MGM.

All subjects reported that the raw data—even with a "small" number of 20 dimensions—was impossible to summarize in a 5 minute period. Subjects also reported that the aggregation of states in the MGM helped them summarize the data faster rather than having to aggregate manually. While none of them explicitly indicated they noticed that all the rows of the MGM were relevant, they did report that it was easier to find the differences. One strictly preferred the MGM over the options, while another found the MGM easier for making up a narrative but was overall satisfied with both the MGM and the K-means clustering. One subject appreciated the succinctness of the MGM but was concerned that "it may lose some information". This final comment motivates future work on structured priors for on what logical formulas should be allowed or likely; future user studies should study the effects of the feature extraction and selection separately. Finally, a qualitative review of the summaries produced found similar but slightly more compact organization of notes in the MGM compared to the K-means model.

# 5 Discussion and Related Work

MGM combines extractive and selective approaches for finding a small set of distinguishable dimensions when performing unsupervised learning on high-dimensional data sets. Rather than rely on criteria that use statistical measures of variation, and then performing additional post-processing to interpret the results, we build interpretable criteria directly into the model. Our logic-based feature extraction step allows us to find natural groupings of dimensions such as (backbone or tail or toothless) in the animal data and (salt or pepper or cream) in the recipe data. Defining an interesting dimension as one whose parameters are drawn from a multi-modal distribution helps us recover groups like pasta and punch. Providing such comprehensive lists of distinguishing dimensions assists in the data exploration and hypothesis generation process. Similarly, providing lists of dimensions that have been consolidated in one extraction aids the human discovery process of why those dimensions might be a meaningful group.

Closest to our work are feature selection approaches such as [17, 18, 19], which also use a mixture of beta-distributions to identify feature types. In particular, [17] uses a similar hierarchy of Beta and Bernoulli priors to identify important dimensions. They carefully choose the priors so that some dimensions can be globally important, while other dimensions can be locally important. The parameters for important dimensions are chosen IID from a Gaussian distribution, while values for all unimportant dimensions come from the *same* background distribution.

Our approach draws parameters for important dimensions from distributions with multiple modes— while unimportant dimensions are drawn from a uni-modal distribution. Thus, our model is more expressive than approaches in which all unimportant dimension values are drawn from the same distribution. It captures the idea that not all variation is important; clusters can vary in their emission parameters for a particular dimension and that variation still might not be interesting. Specifically, an important dimension is one where there is a *gap* between parameter values. Our logic-based feature extraction step collapses the dimensionality further while retaining interpretability.

More broadly, there are many other lines of work that focus on creating latent variable models based on diversity or differences. Methods for inducing diversity, such as determinantal point processes [26], have been used to find diverse solutions on applications ranging from detecting objects in videos [27], topic modeling [28], and variable selection [29]. In these cases, the goal is to avoid finding multiple very similar optima; while the generated solutions are different, the model itself does not provide descriptions of what distinguishes one solution from the rest. Moreover, there may be situations in which forcing solutions to be very different might not make sense: for example, when clustering recipes, it may be very sensible for the ingredient "salt" to be a common feature of all clusters; likewise when clustering patients from an autism cohort, one would expect all patients to have some kind of developmental disorder.

Finally, other approaches focus on building models in which factors describe what distinguishes them from some baseline. For example, [30] builds a topic model in which each topic is described by the difference from some baseline distribution. Contrastive learning [31] focuses on finding the directions that are most distinguish background data from foreground data. Max-margin approaches to topic models [32] try to find topics that can best assist in distinguishing between classes, but are not necessarily readily interpretable themselves.

# 6 Conclusions and Future Work

We presented MGM, an approach for interpretable feature extraction and selection. By incorporating interpretability-based criteria directly into the model design, we found key dimensions that distinguished clusters of animals, recipes, and patients. While this work focused on the clustering of binary data, these ideas could also be applied to mixed and multiple membership models. Similarly, notions of interestingness based on a gap could be applied to categorical and continuous data. It also would be interesting to consider more expressive extracted features, such as more complex logical formulas. Finally, while we learned feature extractions in a completely unsupervised fashion, our generative approach also allows one to flexibly incorporate domain knowledge about possible group memberships into the priors.

## Footnotes

[1]Computer Cooking Contest: http://liris.cnrs.fr/ccc/ccc2014/doku.php

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
