[Supplementary Material]


Supplement material for
# Mind the Gap: A Generative Approach to Interpretable Feature Selection and Extraction

Been Kim              Julie Shah              Finale Doshi−Velez

Massachusetts Institute of Technology,          Harvard University
{beenkim, julie_a_shah}@csail.mit.edu          finale@seas.harvard.edu

## Evidence Lower Bound (ELBO)

Minimizing the Kullback-Leibler divergence between the true posterior $p(\theta|\{w_{nd}\})$ and the variational distribution $q(\theta)$ corresponds to maximizing the lower bound on the evidence $E_q[\log p(\theta|\{w_{nd}\})] - H(q)$ where $H(q)$ is the entropy. We expand the terms in $E_q[\log p(\theta|\{w_{nd}\})] - H(q)$ into

$$
\begin{aligned}
L(t, y, \beta, \pi, z) &= E_q[\log p(\pi|\alpha)] & (1) \\
&+ \sum^{N} E_q[\log p(z_n|\pi)] & (2) \\
&+ \sum^{N}\sum^{D} E_q[\log p(w_{nd}|i_{ng}, f, l_d)] & (3) \\
&+ \sum^{N}\sum^{G} E_q[\log p(i_{ng}|\beta, z_n)] & (4) \\
&+ \sum^{G}\sum^{K} E_q[\log p(\beta_{gk}|t_{gk}, y_g)] & (5) \\
&+ \sum^{G}\sum^{K} E_q[\log p(t_{gk}|\gamma_g)] & (6) \\
&+ \sum^{G} E_q[\log p(\gamma_g|\sigma)] & (7) \\
&+ \sum^{G} E_q[\log p(y_g|\rho)] & (8) \\
&+ \sum^{G} E_q[\log p(f_g|\iota)] & (9) \\
&+ \sum^{D} E_q[\log p(l_d|\kappa) & (10) \\
&- H(q) & (11)
\end{aligned}
$$

and derive the value of each of the terms below.

**1st term**  $\pi$ are the cluster popularities. $\Psi$ is the digamma function.

$$
\begin{aligned}
E_q[\log p(\pi|\alpha)] &= E_q[\log(\pi^{\alpha-1}) - \sum^{K}\log\Gamma(\alpha_i) + \log\Gamma(\sum^{K}\alpha_i)] \\
&= \sum^{K}(\alpha_i - 1)E_q[\log(\pi_k)] - \sum^{K}\log\Gamma(\alpha_i) + \log\Gamma(\sum^{K}\alpha_i) \\
&= \sum^{K}(\alpha_i - 1)(\Psi(\tau_k) - \Psi(\sum_{j}\tau_j)) - \sum^{K}\log\Gamma(\alpha_i) + \log\Gamma(\sum^{K}\alpha_i)
\end{aligned}
$$

**2nd term** $z_n$ encodes the assignments of observations to clusters.

$$
\begin{aligned}
E_q[\log p(z_n|\pi)] &= E_q[\sum_{}^{K} \log(\pi_k^{\mathbf{1}(z_n=k)})] \\
&= \sum_{}^{K} E[\mathbf{1}(z_n=k)]E[\log \pi_k] \\
&= \sum_{}^{K} \nu_{nk} * (\Psi(\tau_k) - \Psi(\sum_j \tau_j))
\end{aligned}
$$

**3rd term** Likelihood.

$$
\begin{aligned}
E_q[\log p(w_{nd}|i_{ng}, f, l_d)] &= E_q[\log(\prod_g \mathbf{1}(l_d=g)[(1)^{\mathbf{1}(f_g=1)\mathbf{1}(i_{ng}=1)\mathbf{1}(w_{nd}=1)}(\mathbf{0})^{\mathbf{1}(f_g=1)\mathbf{1}(i_{ng}=1)\mathbf{1}(w_{nd}=0)} \\
&\qquad (\frac{1}{2})^{\mathbf{1}(f_g=0)\mathbf{1}(i_{ng}=1)}(\mathbf{0})^{\mathbf{1}(i_{ng}=0)\mathbf{1}(w_{nd}=1)}(1)^{\mathbf{1}(i_{ng}=0)\mathbf{1}(w_{nd}=0)} \\
&= \sum_g E_q[\mathbf{1}(l_d=g)\mathbf{1}(f_g=1)\mathbf{1}(i_{ng}=1)\mathbf{1}(w_{nd}=1)\log(1) \\
&\quad +\mathbf{1}(l_d=g)\mathbf{1}(f_g=1)\mathbf{1}(i_{ng}=1)\mathbf{1}(w_{nd}=0)\log(0) \\
&\quad +\mathbf{1}(l_d=g)\mathbf{1}(f_g=0)\mathbf{1}(i_{ng}=1)\log(\frac{1}{2}) \\
&\quad +\mathbf{1}(l_d=g)\mathbf{1}(i_{ng}=0)\mathbf{1}(w_{nd}=1)\log(\mathbf{0}) \\
&\quad +\mathbf{1}(l_d=g)\mathbf{1}(i_{ng}=0)\mathbf{1}(w_{nd}=0)\log(\mathbf{1})
\end{aligned}
$$

**4th term** $i_{ng}$ indicates the presence of group $g$ in observation $n$.

$$
\begin{aligned}
E_q[\log p(i_{ng}|\beta, z_n)] &= E_q[\log(\beta_{gz_n}^{i_{ng}}(1-\beta_{g,z_n})^{1-i_{ng}})] \\
&= i_{ng}E_q[\log \beta_{gz_n}] + (1-i_{ng})E_q[\log(1-\beta_{gz_n})] \\
&= i_{ng}E_q[\sum_{}^{K} \mathbf{1}(\mathbf{z_n}=\mathbf{k})\log \beta_{gk}] + (1-i_{ng})E_q[\sum_{}^{K} \mathbf{1}(\mathbf{z_n}=\mathbf{k})\log(1-\beta_{gk})] \\
&= i_{ng}\sum_{}^{K} E_q[v_{nk}\log \beta_{gk}] + (1-i_{ng})\sum_{}^{K} E_q[\nu_{nk}\log(1-\beta_{gk})] \\
&= o_{ng}\sum_{}^{K} \nu_{nk}(\Psi(\phi_{gk1}) - \Psi(\phi_{gk1}+\phi_{gk2})) + (1-o_{ng})\sum_{}^{K} \nu_{nk}(\Psi(\phi_{gk2}) - \Psi(\phi_{gk1}+\phi_{gk2}))
\end{aligned}
$$

**5th term:** $\beta_{gk}$ is the probability of each group for each cluster.

$$
\begin{aligned}
E_q[\log p(\beta_{gk}|t_{gk}, y_g)] &= E_q[\log(\left(\mathsf{Beta}(\alpha_t, \beta_t)^{t_{gk}}\mathsf{Beta}(\alpha_b, \beta_b)^{(1-t_{gk})}\right)^{y_g}\mathsf{Beta}(\alpha_u, \beta_u)^{1-y_g})]\\
&= E_q[y_g t_{gk}\log\mathsf{Beta}(\alpha_t, \beta_t) + y_g(1-t_{gk})\log\mathsf{Beta}(\alpha_b, \beta_b) + (1-y_g)\mathsf{Beta}(\alpha_u, \beta_u)]\\
&= E_q[y_g t_{gk}\{(\alpha_t - 1)\log(\beta_{gk}) + (\beta_t - 1)\log(1 - \beta_{gk}) - \log\mathsf{BetaFun}(\alpha_t, \beta_t)\}\\
&\quad + y_g(1-t_{gk})\{(\alpha_b - 1)\log(\beta_{gk}) + (\beta_b - 1)\log(1 - \beta_{gk}) - \log\mathsf{BetaFun}(\alpha_b, \beta_b)\}\\
&\quad + (1-y_g)\{(\alpha_u - 1)\log(\beta_{gk}) + (\beta_u - 1)\log(1 - \beta_{gk}) - \log\mathsf{BetaFun}(\alpha_u, \beta_u)]\\
&= E[y_g]E[t_{gk}]\{(\alpha_t - 1)E[\log(\beta_{gk})] + (\beta_t - 1)E[\log(1 - \beta_{gk})] - \log\mathsf{BetaFun}(\alpha_t, \beta_t)\}\\
&\quad + E[y_g](1 - E[t_{gk}])\{(\alpha_b - 1)E[\log(\beta_{gk})] + (\beta_b - 1)E[\log(1 - \beta_{gk})] - \log\mathsf{BetaFun}(\alpha_b, \beta_b)\}\\
&\quad + (1 - E[y_g])\{(\alpha_u - 1)E[\log(\beta_{gk})] + (\beta_u - 1)E[\log(1 - \beta_{gk})] - \log\mathsf{BetaFun}(\alpha_u, \beta_u)\}\\
&= E[\log(\beta_{gk})]\left(E[y_g]E[t_{gk}](\alpha_t - 1) + E[y_g](1 - E[t_{gk}])(\alpha_b - 1) + (1 - E[y_g])(\alpha_u - 1)\right)\\
&\quad + E[\log(1 - \beta_{gk})]\left(E[y_g]E[t_{gk}](\beta_t - 1) + E[y_g](1 - E[t_{gk}])(\beta_b - 1) + (1 - E[y_g])(\beta_u - 1)\right)\\
&\quad - E[y_g]E[t_{gk}]\log\mathsf{BetaFun}(\alpha_t, \beta_t)\\
&\quad - E[y_g](1 - E[t_{gk}])\log\mathsf{BetaFun}(\alpha_b, \beta_b)\\
&\quad - (1 - E[y_g])\log\mathsf{BetaFun}(\alpha_u, \beta_u)\\
&= (\Psi(\phi_{gk1}) - \Psi(\phi_{gk1} + \phi_{gk2}))\left(\eta_g\lambda_{gk}(\alpha_t - 1) + \eta_g(1 - \lambda_{gk})(\alpha_b - 1) + (1 - \eta_g)(\alpha_u - 1)\right)\\
&\quad + (\Psi(\phi_{gk2}) - \Psi(\phi_{gk1} + \phi_{gk2}))\left(\eta_g\lambda_{gk}(\beta_t - 1) + \eta_d(1 - \lambda_{gk})(\beta_b - 1) + (1 - \eta_g)(\beta_u - 1)\right)\\
&\quad - \eta_g\lambda_{gk}\log\mathsf{BetaFun}(\alpha_t, \beta_t)\\
&\quad - \eta_g(1 - \lambda_{gk})\log\mathsf{BetaFun}(\alpha_b, \beta_b)\\
&\quad - (1 - \eta_g)\log\mathsf{BetaFun}(\alpha_u, \beta_u)
\end{aligned}
$$

**6th term:** $t_g$ is the auxiliary variable indicating from which mode an important $\beta_{gk}$ was drawn.

$$
\begin{aligned}
E_q[\log p(t_{gk}|\gamma_g) &= E_q[\log(\gamma_g^{t_{gk}}(1 - \gamma_g)^{1-t_{gk}}]\\
&= E[t_{gk}]E[\log\gamma_g] + (1 - E[t_{gk}])E[\log(1 - \gamma_g)]\\
&= \lambda_{gk}(\Psi(\ell_{g1}) - \Psi(\ell_{g1} + \ell_{g2})) + (1 - \lambda_{gk})(\Psi(\ell_{g2}) - \Psi(\ell_{g1} + \ell_{g2}))
\end{aligned}
$$

**7th term:** $\gamma_g$ is the proportion of each mode in multimodal distribution.

$$
\begin{aligned}
E_q[\log p(\gamma_g|\sigma)] &= E_q[\log p(\gamma_g|\sigma)]\\
&= E_q[\log(\frac{\gamma_g^{\sigma_1 - 1}(1 - \gamma_g)^{\sigma_2 - 1}}{\mathsf{BetaFun}(\sigma_1, \sigma_2)})]\\
&= E_q[\log\left(\frac{\gamma_g^{\sigma_1 - 1}(1 - \gamma_g)^{\sigma_2 - 1}}{\mathsf{BetaFun}(\sigma_1, \sigma_2)}\right)] + \mathrm{Const}\\
&= ((\sigma_1 - 1)E[\log\gamma_g] + (\sigma_2 - 1)E[\log(1 - \gamma_g)] - \log\mathsf{BetaFun}(\sigma_1, \sigma_2))\\
&= [(\sigma_1 - 1)(\Psi(\ell_{g1}) - \Psi(\ell_{g1} + \ell_{g2}))\\
&\quad + (\sigma_2 - 1)(\Psi(\ell_{g2}) - \Psi(\ell_{g1} + \ell_{g2}))\\
&\quad - \log\mathsf{BetaFun}(\sigma_1, \sigma_2)]
\end{aligned}
$$

**8th term:** $y_g$ indicates whether a dimension is important.

$$
\begin{aligned}
E_q[\log p(y_g)] &= E_q[\log\rho^{y_g}(1 - \rho)^{1-y_g}]\\
&= E[y_g]\log\rho + (1 - E[y_g])\log(1 - \rho)\\
&= \eta_g\log\rho + (1 - \eta_g)\log(1 - \rho)
\end{aligned}
$$

**9th term:** $f_g$ indicates which formula is associated with group $g$.

$$
\begin{aligned}
E_q[\log p(f_g|\iota)] &= E_q[\log\iota^{f_g}(1 - \iota)^{1-f_g}]\\
&= E[f_g]\log\iota + (1 - E[f_g])\log(1 - \iota)\\
&= e_g\log\iota + (1 - e_g)\log(1 - \iota)
\end{aligned}
$$

**10th term:** $l_d$ indicates to which group dimension $d$ belongs.

$$
\begin{aligned}
E_q[\log p(l_d|\kappa)] &= E_q[\sum^{G} \log(\kappa_g^{\mathbf{1}(l_d=g)})] \\
&= \sum^{G} E[\mathbf{1}(l_d = g)]E[\log \kappa_g] \\
&= \sum^{G} c_{dg} * (\Psi(h_g) - \Psi(\sum_g h_g))
\end{aligned}
$$

$$
E_q[\log p(l_d|\kappa)] = 4
$$

$$
= \sum^{G} E[\mathbf{1}(l_d = g)]E[\log \kappa_g]
$$

**11th term:** Entropy term.

$$H(q) = E_q[\log(q(t)q(y)q(\pi)q(\beta)q(z)q(\gamma)q(i)q(l)q(\theta))]$$

$$= E_q[\sum_g^G \sum_k^K \log q(t_{gk}) + \sum_g^G \log q(y_g) + \log q(\pi)$$

$$+ \sum_g^G \sum_k^K \log q(\beta_{gk}) + \sum_n^N \sum_k^K \log q(z_{nk}) + \sum_g^G \log q(\gamma_g) + \sum_n^N \sum_g^G \log q(i_{ng}) + \sum_g^G \log q(\theta_g) + \log p(\kappa)]$$

$$= \sum_g^G \sum_k^K E_q[t_{gk}]\log\lambda_{gk} + (1 - E_q[t_{gk}])\log(1 - \lambda_{gk})$$

$$+ \sum_g^G E_q[y_g]\log(\eta_g) + (1 - E_q[y_g])\log(1 - \eta_g)$$

$$+ \sum_k^K (\tau_k - 1)E_q[\log(\pi_k)] - \sum^K \log\Gamma(\tau_i) + \log\Gamma(\sum^K \tau_i)$$

$$+ \sum_g^G \sum_k^K ((\phi_{gk1} - 1)E[\log(\beta_{gk})] + (\phi_{gk2} - 1)E[\log(1 - \beta_{gk})] - \log BetaFun(\phi_{gk1}, \phi_{gk2}))$$

$$+ \sum_n^N \sum_k^K \nu_{nk}\log v_{nk}$$

$$+ \sum_g^G (\ell_{g1} - 1)E[\log(\gamma_g)] + (\ell_{g2} - 1)E[\log(1 - \gamma_g)] - \log BetaFun(\ell_{g1}, \ell_{g2})$$

$$+ \sum_n^N \sum_g^G E_q[i_{ng}]\log(o_{ng}) + (1 - E_q[i_{ng}])\log(1 - o_{ng})$$

$$+ \sum_d^D \sum_g^G c_{dg}\log c_{dg}$$

$$+ \sum_g^G E_q[f_g]\log(e_g) + (1 - E_q[f_g])\log(1 - e_g)$$

$$= \sum_g^G \sum_k^K \lambda_{gk}\log\lambda_{gk} + (1 - \lambda_{gk})\log(1 - \lambda_{gk})$$

$$+ \sum_g^G \eta_g\log(\eta_g) + (1 - \eta_g)\log(1 - \eta_g)$$

$$+ \sum_k^K (\tau_k - 1)(\Psi(\tau_k) - \Psi(\sum_j^K \tau_j)) - \sum^K \log\Gamma(\tau_i) + \log\Gamma(\sum^K \tau_i)$$

$$+ \sum_g^G \sum_k^K ((\phi_{gk1} - 1)(\Psi(\phi_{gk1}) - \Psi(\phi_{gk1} + \phi_{gk2})) + (\phi_{gk2} - 1)(\Psi(\phi_{gk2}) - \Psi(\phi_{gk1} + \phi_{gk2})) - \log BetaFun(\phi_{gk1}, \phi_{gk2}))$$

$$+ \sum_n^N \sum_k^K \nu_{nk}\log v_{nk}$$

$$+ \sum_g^G (\ell_{g1} - 1)(\Psi(\ell_{g1}) - \Psi(\ell_{g1} + \ell_{g2})) + (\ell_{g2} - 1)(\Psi(\ell_{g2}) - \Psi(\ell_{g1} + \ell_2)) - \log BetaFun(\ell_{g1}, \ell_{g2})$$

$$+ \sum_n^N \sum_g^G o_{ng}\log(o_{ng}) + (1 - o_{ng})\log(1 - o_{ng})$$

$$+ \sum_d^D \sum_g^G c_{dg}\log c_{dg}$$

$$+ \sum_g^G e\log(e_g) + (1 - e_g)\log(1 - e_g)$$