[Reviews · NeurIPS 2015]

Submitted by Assigned_Reviewer_1

This paper proposes a new approach for feature extraction and selection. They first divide the features into non-overlapping groups via logic-based feature extraction technique. Then, they identify important groups as the ones whose parameter values vary a lot across clusters. The idea is interesting and experiments show it matches or outperforms other algorithms. My major concerns are the requirement of binary data and the scalability to large-scale problems, which may limit the practical usage of the proposed approach.

Quality: The idea is interesting and the techniques are sound. My concerns are as follows.

1. As the authors mentioned in the paper, the proposed approach focuses on binary

data. It is unclear if the approach can be easily extended to general design matrix. In Section 4.1, the authors binarized the image data. I think this may not be fair to other algorithms that do not require binary data. Moreover, did you binarize the three data sets in Section 4.2 as well?

2. The dimension of the real-world data sets in Section 4.2 are small. The authors may consider to evaluate the proposed approaches on high-dimensional data sets.

3. Do the proposed method learn the number of clusters like HFS or assume it is given?

Clarity: The paper is mostly accessible. I would suggest the authors to move Figure 2 to the beginning of Section 2 and use it as an example to better illustrate the idea.

Originality: The key idea, i.e., using gap in parameter values to identify important groups, is interesting.

Significance: The idea is interesting and experiments have shown its promising performance on several real-world data sets. However, some issues as I mentioned in 1,2, and 3, may need to be properly addressed.
Summary: This paper proposes a new approach for feature extraction and selection. They first divide the features into non-overlapping groups via logic-based feature extraction technique. Then, they identify important groups as the ones whose parameter values vary a lot across clusters. The idea is interesting and experiments show it matches or outperforms other algorithms. My major concerns are the requirement of binary data and the scalability to large-scale problems, which may limit the practical usage of the proposed approach.

Submitted by Assigned_Reviewer_2

(Light Review) This paper presents a clustering method that

purports to provide greater interpretability for what

distinguishes one cluster from another.

The authors do this by

first providing a feature selection method that creates logical

rules to group sets of features that vary together, and second

by modeling distinguishing parameters as having a bi-modal

distribution (i.e., with a gap in feature values that

distinguishes different clusters) vs. non-distinguishing

parameters as having a unimodal distribution (possibly of high

variance).

These choices are well motivated and the development

of the technique is relatively clear; the work also appears to

be novel in its approach.

Results are shown first in terms of

mutual information with underlying labeled clusters, for which

it performs in the top two for the set of baselines presented in

both data scenarios.

However, the main point of the technique

(and the paper) was greater interpretabilty, and the evaluation

for this is very weak - the authors essentially talk about the

features that were found as distinguishing and explain that

they're reasonable.

While their explanations are sensible,

they're ultimately insufficient to justify the premise of the

paper, as (a) such a subjective judgment requires multiple

judgments by human subjects (via Turk, etc.) rather than

arguments by the authors, and (b) no baseline for comparison is

shown, while in fact there are other methods that purport to

describe the differences between clusters (albeit focused on the

most discriminating features, as the authors point out).

Since

this was presented as the primary contribution of the paper, it

seems that the authors have not sufficiently demonstrated that

their method has satisfied this claim.

Summary: This paper presents a clustering method that purports to provide

greater interpretability for what distinguishes one cluster from

another. The method seems sensible and the results (in terms of

MI with labeled clusters) are competitive, but there is no

evaluation for the interpretability of the results (the claimed

core contribution of the paper) beyond the authors' arguing that

they are reasonable.

Submitted by Assigned_Reviewer_3

The paper proposed a new model for feature selection and extraction. The key is to first group features then do feature selection or feature extraction. The formulation of the model is given on page 3. The problem studied in the paper is of great significance and the idea presented in the paper is also interesting. Below are my concerns on the paper: 1. Although the model proposed in the paper is new, the idea is not. Grouping features first then selecting features is a well-known approach in analytics industry. For instance, see PROC VARCLUS: http://support.sas.com/documentation/cdl/en/statug/63033/HTML/default/viewer.htm#statug_varclus_sect001.htm. People usually use this procedure to group and select features. 2. The authors need to provide time complexity analysis for the proposed algorithm. 3. The authors tested different algorithms on only two data sets and none of them are of very high dimensionality.
Summary: The paper proposed a new model for feature selection and extraction. The key is to first group features then do feature selection or feature extraction. The formulation of the model is given on page 3.

Author Feedback
Author rebuttal: We first would like to thank the reviewers for thoughtful comments.

** Reviewer 3/Novelty: The idea of extraction then selection is not new. We also noted this in [8], and we will include a references to VARCLUS in final version. That said, our approach has important novel aspects:

1. Our combination of logic-based extraction and gap-based feature selection is aimed toward interpretability, not just dimensionality reduction. For example, the VARCLUS method uses harder-to-interpret linear combinations of cluster components as features.

2. Our model creates a single objective function for the whole procedure - rather than using one metric for dimensionality reduction and then something completely different for variable selection.

** Reviewer 6/Human Subject Experiments. We agree that evaluating interpretability is hard, and human subject experiments provide a way for evaluation. We are currently getting feedback from the domain experts who provided the disease data and will include that in the final version.

** Reviewer 4/Quantitative Results. Table 1 in the paper provides quantitative baseline comparisons with a number of feature selection methods mentions in [17]. This includes Kmeans, HFS(G) (Guan et al. 2011), Law (Law et al. 2002), DPM (simple Dirichlet process mixture of Gaussians), HFS(L) and Cc (Cho et al. 2004). Our method performs at the/among the top for the set of baselines.

** Reviewer 1/Binary Data. Discrete data can be easily binarized, and continuous data, once passed through logical expressions in the extraction stage, would also become binary. Thus, we believe this is not a severe restriction.

** Reviewers 1, 3/Dimensionality. The inference scales in the number of extracted dimensions, which is domain-dependent and data-driven and might be smaller than the total number of dimensions as groups are created. Our gap-based approach adds very little overhead per dimension over standard clustering approaches -- in particular unimportant dimensions are just like standard clustering models-- and so that part also scales. We also note that interpreting even modestly high-dimensional data--such as data sets with 100s of dimensions--is nontrivial.

** Reviewer 3/Time Complexity. We will provide time complexity in the final version.

** Reviewer 1/Figure 2. Moving figure 2 to section 2 is a great idea - we will do this for the camera ready version.